# ENHANCING GRAPH OF THOUGHT: ENHANCING PROMPTS WITH LLM RATIONALES AND DYNAMIC TEMPERATURE CONTROL

**Sunguk Shin and Youngjoon Kim**[*]
Korea University
Seoul, Republic of Korea
{ssw1419, acorn421}@korea.ac.kr

## ABSTRACT

We introduce Enhancing Graph of Thoughts (EGoT), a method designed to enhance the performance of large language models (LLMs) on complex reasoning tasks. EGoT automates the process of generating accurate responses using given data and a base prompt. The process consists of several steps: It obtains an initial response from the answering node using the base prompt. Evaluation node evaluates the response and generates reasoning for it, utilizing the score's probabilities to enhance evaluation accuracy. The reasoning from both the answering node and the evaluation node is aggregated to identify the problem in the response. This aggregated reasoning is incorporated into the base prompt to obtain an enhanced response. These steps are organized in a graph architecture, where the final leaf nodes are merged to produce a final response. As the graph descends, the temperature is lowered using Cosine Annealing and scoring, to explore diverse responses with earlier nodes and to focus on precise responses with later nodes. The minimum temperature in Cosine Annealing is adjusted based on scoring, ensuring that nodes with low scores continue to explore diverse responses, while those with high scores confirm accurate responses. In sorting 256 elements using GPT-4o mini, EGoT performs 88.31% accuracy, while GoT (Graph of Thoughts) achieves 84.37% accuracy. In the frozen lake problem using GPT-4o, EGoT averages 0.55 jumps or falls into the hole, while ToT (Tree of Thoughts) averages 0.89.

## 1 INTRODUCTION

In recent research, the performance of large language models (LLMs) has evolved incredibly rapidly, with applications in a variety of fields, including math problem (Shao et al., 2024), robotics (Park et al., 2023), medicine (Lee et al., 2024b; Kwon et al., 2024), and even programming (Wang et al., 2023a; Duong & Meng, 2024; McAleese et al., 2024). To further improve the performance of LLMs, researchers are now actively exploring methods to significantly scale up the architecture of models, or optimize models through distillation (Qu et al., 2024) and fine-tuning (Singh et al., 2024). These efforts are broadening the scope of LLMs and enabling more innovative applications.

Training LLMs directly requires significant time and GPU resources. To address such limitations, Prompt engineering, which involves designing effective prompts rather than training the model directly, stands out. Prompt engineering is a technique that can improve the performance of LLMs on specific tasks without requiring additional training. Examples of prompt engineering include Chain of Thought (CoT) (Wei et al., 2022), Chain of Thought with Self-Consistency (CoT-SC) (Wang et al., 2023b), Tree of Thoughts (ToT) (Long, 2023; Yao et al., 2024), Exchange of Thought (EoT) (Yin et al., 2023), and Graph of Thoughts (GoT) (Besta et al., 2024). These approaches help LLMs generate more accurate and useful results.

However, complex problems often impair the reasoning ability of LLMs. When an LLM provides a correct answer, its rationale steps are not always reliable (Hao et al., 2024). In addition, most architectures utilize external tools (Stechly et al., 2023; Gou et al., 2024) to improve performance,

---

[*]Corresponding author

and prompts often require specific examples (Lee et al., 2024a). Since obtaining the valid rationale makes LLM's performance highly contributing (Yin et al., 2024), the technique of prompting LLMs with a score to evaluate the performance (Valmeekam et al., 2023; Ren et al., 2023) is an ongoing research area. There is also research exploring dynamic temperature control techniques (Cai et al., 2024; Nasir et al., 2024; Zhang et al., 2024; Zhu et al., 2024) to further enhance the reasoning ability of LLMs.

Our approach, EGoT, is an architecture that can automatically generate the prompts and answers from the LLM by only initializing the base prompt. During this process, log probability is utilized to evaluate the LLM's responses, increasing their confidence. We also propose dynamically adjusting the temperature based on the progress and score of the answer, applying the cosine annealing (Loshchilov & Hutter, 2016) to set a high temperature at the beginning of the graph and a low temperature at the end. The minimum temperature is set as the inverse of the score, so that nodes with high scores consistently provide correct answers, while nodes with low scores explore a wide range of answers. This approach has the advantage of showing constant and consistent performance without the evaluation metric, and it does not need additional examples to avoid bias in the results. Note that our study proposes a framework that strategically resolves conflicts from naively merging prior methods through trade-offs, thereby ensuring high performance.

To summarize, EGoT provides the following advantages:

- Dynamic temperature control using Cosine Annealing to propagate more accurate rationales to child node prompts.
- Continuously appending of rationales to the base prompt in graph architecture to generate a high-quality final response.
- Enhanced confidence by utilizing the probability of LLM answers for scoring, while avoiding bias by excluding specific examples.
- Direct repetition of the input question in its original form to improve LLM comprehension, followed by integration of prior repetitions into the rationale.

## 2 EGoT ARCHITECTURE

### 2.1 OVERVIEW

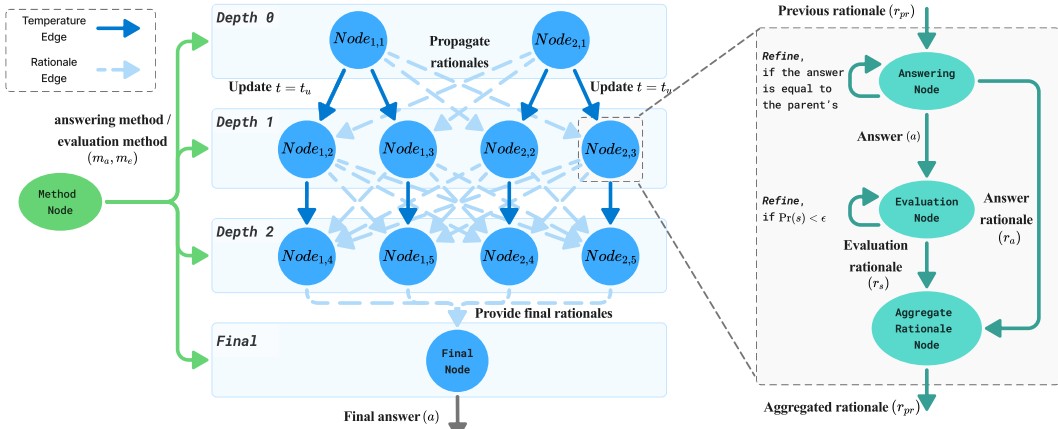

Figure 1: Framework of EGoT. The left side illustrates the overall graph architecture and dynamic temperature. The right side illustrates the internals of each Node. Each Node contains ANSWER-INGNODE, EVALUATIONNODE, and AGGREGATERATIONALENODE as sub-nodes. The temperature parameter updates its child nodes within the tree, propagating the rationale information to deeper levels. As the graph progresses, the temperature decreases and propagates the rationale information.

The EGoT graph structure is shown in Figure 1. Each node consists of three stages: Stage 1 (ANSWERINGNODE) obtains the answer to resolve the problem from the LLM; Stage 2

(EVALUATIONNODE) asks the LLM to evaluate its response and assign a score; and Stage 3 (AGGREGATERATIONALENODE) collects the LLM's rationales from both Stage 1 and Stage 2, forwarding them to the next nodes. METHODNODE is executed only once at the beginning of the overall structure, and this step can be replaced with an expert's problem-solving approach.

## 2.2 METHOD NODE

The METHODNODE inquires about the method for solving the problem and the methods for evaluating the answer. Although these methodologies can be formulated by human experts, in this paper, heuristic methods are requested from the LLM and utilized. $m_a$ denotes the method for obtaining the answer to the question, and $m_e$ denotes the method for evaluating the answer. $t$ denotes the temperature of the LLM.

$$m_a, m_e = \text{METHODNODE}(Prompt, t = 0) \tag{1}$$

## 2.3 ANSWERING NODE

ANSWERINGNODE finds the answer to the problem. The top root node solves the problem with the rules. The child node solves the problem using the rationale from the previous nodes. ANSWERINGNODE outputs the answer to the problem and the rationale for the answer. $a$ and $r_a$ are the answer and the rationale regarding the response provided by the LLM, and $r_{pr}$ denotes the rationales of the previous nodes. In this study, the temperature $t$ is fixed at 1 for the root node, while for all other nodes it is determined by the parent's temperature using cosine annealing. We denote the updated temperature as $t_u$.

$$a, r_a = \begin{cases} \text{ANSWERINGNODE}(Prompt(m_a, \cdot), t = 1), & \text{if} \quad \text{Node = Root Node} \\ \text{ANSWERINGNODE}(Prompt(m_a, r_{pr}), t = t_u), & \text{else} \quad \text{Node} \neq \text{Root Node} \end{cases} \tag{2}$$

## 2.4 EVALUATION NODE

EVALUATIONNODE evaluates the answer provided by ANSWERINGNODE. The LLM outputs both the accuracy of the answer and the rationale for that accuracy score. If the probability of the score provided by the LLM is lower than the threshold, EVALUATIONNODE is executed again. $s$ and $r_s$ are the score and the rationale regarding the LLM's response, and $\Pr(s)$ is the probability of the score. We request a score range of 0 to 100 from the LLM to better represent the scores as percentages.

$$s, r_s, \Pr(s) = \text{EVALUATIONNODE}(Prompt(m_e, a), t = 0) \tag{3}$$

## 2.5 AGGREGATE RATIONALE NODE

AGGREGATERATIONALENODE integrates the rationales provided from ANSWERINGNODE and EVALUATIONNODE. The LLM outputs the aggregated rationale along with the information considered inaccurate. AGGREGATERATIONALENODE aggregates the information from the answer rationale and the evaluation rationale, emphasizing the incorrect encountered during the reasoning while omitting details related to successful outputs. This concept is similar to the state evaluator in ToT (Yao et al., 2024); however, our approach provides a rationale for identifying flaws without the answer. The inaccurate information refers to elements that the LLM needs to recheck when conflicts occur between the two input rationales. It arises from the LLM's misinterpretation of the problem and can lead to hallucinations and incorrect reasoning. This information, derived from AGGREGATERATIONALENODE, is subsequently incorporated into the prompt of the child's ANSWERINGNODE. $r_{pr}$ denotes the aggregated rationale and the inaccurate information.

$$r_{pr} = \text{AGGREGATERATIONALENODE}(Prompt(r_a, r_s), t = 0) \tag{4}$$

## 3 METHODOLOGY

### 3.1 ENHANCING RESPONSE

This section describes the methods to obtain enhancing responses from the LLM. Two main approaches are used: exploring varied answers for obtaining enriched responses and utilizing the probability of the answers for more accurate scoring.

#### 3.1.1 EXPLORING VARIED ANSWER

To explore different answers, multiple root nodes are utilized in the architecture. Since the temperature decreases along the node depth, multiple graphs are used to generate different answers. In some cases, a node provides the same answer as its parent. To address this, if a child ANSWERINGNODE gives the same answer as its parent ANSWERINGNODE, a question is only asked once more. This is because it cannot be determined exactly whether it is the correct answer during the entire process of the graph.

#### 3.1.2 ENHANCING SCORE

To enhance the scoring process, the probability that the LLM predicts the score token is used to answer the score. If the probability does not exceed a predefined threshold, the LLM is prompted for the score again. If the LLM outputs extreme score values, such as 0 or 100, a higher threshold is applied because these extreme scores are considered reliable only when the LLM is highly confident. For scores ranging from 1 to 99, the threshold is set lower to filter out nonsensical answers. It is important to consider the order in which the LLM is asked for the score and the rationale for the score. If the LLM is asked for the rationale first and then the score, the LLM thinks that it has a basis in the previous rationale. Therefore, a score of 0 or 100 is often returned regardless of whether the answer is correct or not, with a probability close to 1. For this reason, the score is asked for before the rationale, and the score is obtained with a variety of scores. Detailed explanations of the threshold settings are provided in each experiment.

### 3.2 TEMPERATURE CONTROL

The temperature in LLMs is typically set to 1.0 when generating creative answers. Whereas when creativity is not required, the temperature is set closer to 0 for consistent answers. However, setting the temperature to 0 from the start can lead to fixed answers and errors.

To gradually decrease temperature as the graph progresses, we employ cosine annealing. When a high-quality answer is generated, the temperature is reduced to produce a fixed response, whereas when the answer is uncertain, the temperature is kept high to explore different answers. The purpose for evaluating answers in EVALUATIONNODE is not only to generate a rationale but also to control the temperature. If the score is high, it indicates that the rationale of that ANSWERINGNODE is correct. Therefore, this rationale is forwarded to the child nodes, which are expected to generate good answers. On the other hand, if the score is low, the answer needs to be revised, and the rationale of ANSWERINGNODE also needs to improve, requiring various explorations until it is correct.

In cosine annealing, the maximum temperature ($t_{max}$) is fixed at 0.7 and the minimum temperature is set to the inverse of the accuracy. This means that higher accuracy results in a lower temperature. The total epoch is set to the total number of nodes ($N_t$) and the current epoch is defined as the progress of the nodes ($N_c$).

$$t_u = t_{min} + \frac{1}{2}(t_{max} - t_{min})(1 + cos(\frac{N_c}{N_t})), t_{min} = 1 - \sqrt{1 - (c - 1)^2}, c = s \cdot \Pr(s)^{\frac{1}{e}} \quad (5)$$

Here, $c$ represents the confidence of ANSWERINGNODE. If an answer receives a high score and the probability assigned to that score by the LLM is also high, the confidence is high. Conversely, if an answer receives a low score or the probability assigned to the score is low, the confidence is low. $c$ and $t_{min}$ are between 0 and 1. The probability is used in $t_{min}$ to differentiate between high and low probability cases when the LLM answers the score.

## 3.3 EXAMPLE USE CASE

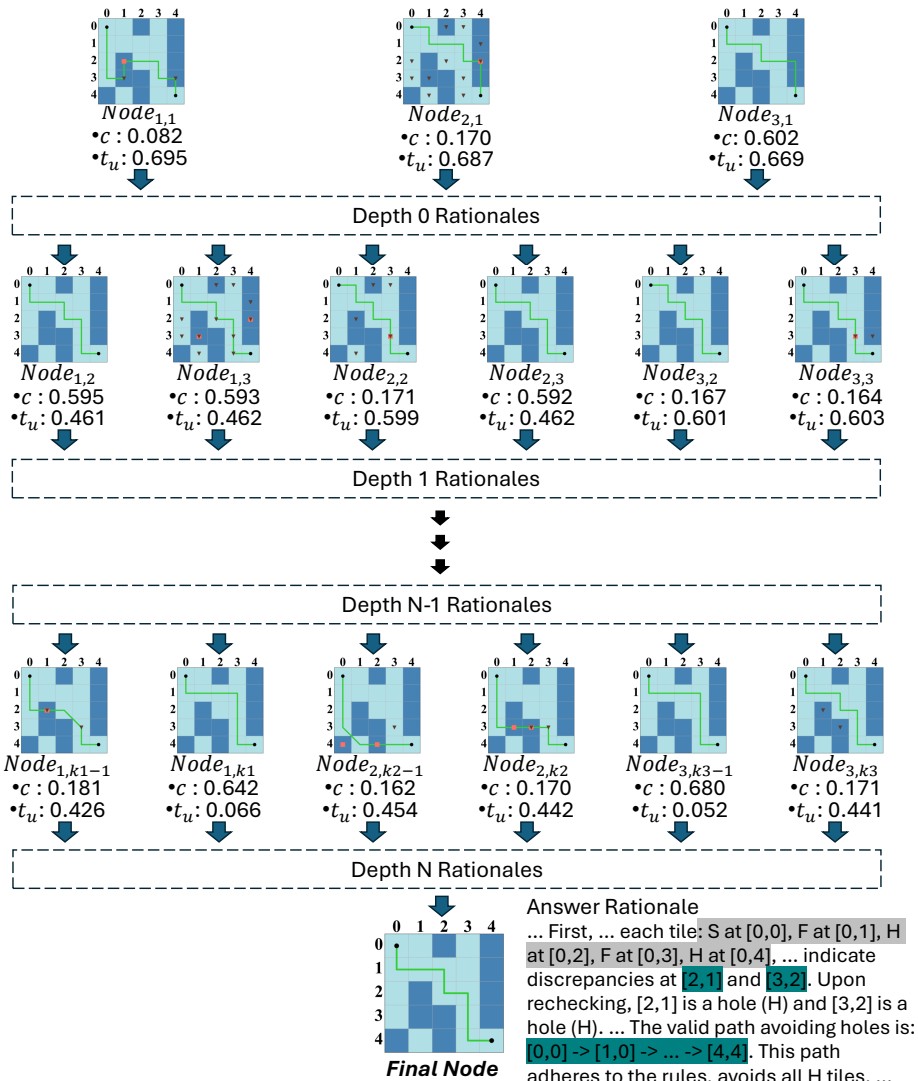

Figure 2: In the Frozen Lake example, the temperature decreases as it progresses down the graph, various positions are explored and the graph finds the correct answer using the information.

This section uses a practical example to illustrate the approaches presented in Sections 2 and 3. Figure 2 shows the results of the Frozen Lake experiment, one of the experimental results that demonstrate the advantages of EGoT. The blue background represents the hole and the light blue represents the frozen tile. The two black points on the top left $(0, 0)$ and bottom right $(4, 4)$ represent the start and end points. The green line indicates the route that the LLM predicts as the answer, the orange square marks what EVALUATIONNODE rationale explains as incorrect because it is a hole. The brown triangle represents the position that AGGREGATERATIONALENODE aggregates because the rationale from ANSWERINGNODE and EVALUATIONNODE conflict with each other.

Before the graph starts, METHODNODE is invoked once. The information provided by the METHODNODE is utilized by all subsequent nodes in Figure 2, from $Node_{1,1}$ to the Final Node. In this experiment, the graph starts with 3 root nodes. In ANSWERINGNODE, $Node_{1,1}$ passes through the holes $(2, 1)$, $(3, 1)$, and $Node_{2,1}$ and $Node_{3,1}$ pass through the holes $(2, 4)$, $(3, 4)$. At EVALUATIONNODE, $Node_{1,1}$ observes the hole at $(2, 1)$ and $Node_{2,1}$ observes the hole at $(2, 4)$. As a result, ANSWERINGNODE states that the answer is incorrect, lowering the confidence of $Node_{1,1}$

and $Node_{2,1}$. Conversely, $Node_{3,1}$ has high confidence in EVALUATIONNODE, because it does not find anything wrong. Since it is the first round, the temperature remains close to 0.7, regardless of confidence. The node updates the temperature of its two child nodes. $Node_{1,2}$ and $Node_{1,3}$ update the temperature by $Node_{1,1}$.

Because depth 0 informs that the coordinates (2, 1) and (2, 4) are holes, depth 1 nodes recognize them as holes and do not traverse these coordinates. Still, $Node_{1,3}$, $Node_{2,2}$, and $Node_{3,3}$ are unsure of the correct answer because the propagated rationale confuses the information about frozen tile and hole. The nodes in depth 1 also cannot make a confident decision and incorrectly state that (3, 3) is a hole. Since one depth has passed, nodes with higher confidence have a lower temperature to update to their child. In the middle of the process (the omitted part of the figure), if a node gives an incorrect answer, the temperature increases again, and it explores the coordinate (3, 2). When the final node responds to the answer by incorporating aggregate rationales from the leaf nodes, the LLM explores the correct answer, avoiding (2, 1) and (3, 2).

## 4 EXPERIMENTS

We use the LangChain (Chase, 2022) library to construct the graph. The graph structure starts with three root nodes, and when solving a problem, the LLM responds with prompts that include all rationale information from the previous depth. At the end of the graph, the answer is aggregated into one by using the response from ANSWERINGNODE with the prompt that incorporates all the aggregated rationales from the leaf nodes.

EGoT is evaluated through three experiments: document merging, number sorting, and Frozen Lake. In the document merging and number sorting experiments, we use the graphs with a depth of 3, and we use a graph with a depth of 4 in the Frozen Lake experiment. We experiment with ToT (Long, 2023) which appends the incorrect answer rather than evaluating and exploring each element. This is due to, in the experiments, the number of nodes increases exponentially to explore each case. In the original paper, GoT selects the best-performing node to evaluate the graph; however, it has been modified to select a median value to compare structural performance alone. Solving problems with evaluation metrics is not considered a structural advantage. Therefore, to fully automate the LLM process, the evaluation of nodes is assumed to be randomized and the median value is used as the expectation. Experiments are conducted multiple times with the same data. To compare the impact of temperature, the experiment is conducted with a temperature fixed to 1, referred to as EGoT*.

### 4.1 DOCUMENT MERGING

We conducted an experiment with the dataset and the evaluation prompt provided by GoT for document merging. The evaluation compares non-redundancy and retained harmonic mean. The performance scores for each method are as follows: IO (75.96%), **CoT (77.79%), ToT (76.74%)**, GoT (76.43%), EGoT (76.01%), and EGoT* (74.98%). This experiment suggests that scoring with an LLM should not simply be evaluated. The experiment shows that autonomous evaluation by an LLM does not have the logical and structural advantages of well-known CoT and ToT. Furthermore, it supports the idea that scoring should be evaluated more rigorously.

### 4.2 NUMBER SORTING

This experiment involves a sorting problem with random numbers as input. The LLM is able to sort short lists successfully. However, its performance decreases when sorting longer lists of numbers. To evaluate the sorting problem, two metrics are utilized: accuracy and number of errors (NOE). Accuracy is calculated as the intersection divided by the union to measure how similar the final result is to the ground truth. The number of errors represents the number of elements that ascend rather than descend. The higher the accuracy, the better, the lower the number of errors, the better. All nodes except ANSWERINGNODE set the temperature to 0. The threshold for score probability is set to 0.99 for 100 and 0, and 0.5 for others.

The experiment uses 100 lists of 128 elements and 100 lists of 256 elements. For the 128-element lists, numbers are randomly selected from the range 1 to 1000, allowing for duplicates. For the 256-element lists, numbers are randomly selected from the range 1 to 1500, also allowing for du-

plicates, because GPT-4o's tokenizer splits numbers over 1000 into two tokens. In this experiment, to demonstrate the effectiveness of repeating the question, CoT is performed in two ways. CoT1 utilizes the rationale to sort the entire list in three steps: divide the list into four parts, sort each part, and then combine them. CoT2 involves rewriting the input to ensure better understanding before sorting the corresponding numbers.

### 4.3  FROZEN LAKE

A Frozen Lake is a problem of finding a route to a destination while avoiding the holes. To find the correct route in a frozen lake, it is necessary to know the exact locations of the holes and understand the rules of the Frozen Lake. To evaluate the Frozen Lake problem, two metrics are utilized: accuracy and number of errors (NOE). Accuracy is the number of successful routes found correctly divided by the total number of attempts. The number of errors is defined as the sum of the number of times the agent falls into a hole and the distance of the jump, which is not valid in the problem setting. All nodes except ANSWERINGNODE set the temperature to 0. The threshold for score probability is set to 0.95 for 100 and 0, and 0.5 for other scores. This experiment is conducted on a 5 by 5 size lake with 20 test cases containing 8 holes and 20 test cases containing 10 holes. Both GPT-4o and GPT-4o mini are used in the experiment.

## 5  EVALUATION

### 5.1  NUMBER SORTING

Table 1 presents the experimental result of number sorting. ToT achieves the best performance when sorting 128 elements, followed by the proposed EGoT. When sorting 256 elements, the proposed EGoT outperforms the other architectures. EGoT also achieves performance similar to that of EGoT, although slightly lower. Note that five experiments were conducted to verify the consistent performance of EGoT for 128 elements and 256 elements. The results are shown in Figure 3, which demonstrates generally consistent performance.

The result of CoT1 and CoT2 is the one to focus on here. While there is a relatively slight performance difference when sorting 128 numbers, the performance gap is significantly larger when sorting 256 numbers. The reason for the difference is that in the first step of CoT1's rationale, when dividing the list into 4 lists, many numbers are missing, and in the last step of the rationale, when merging the 4 lists, it sometimes returns only the numbers from the first list without merging. For this reason, the performance of CoT1 is significantly lower compared to the other experiments. Conversely, CoT2's first step of rationale, which is to repeat elements once more, is relatively simple for the LLM, leading to fewer missing numbers. Subsequently, when prompting for sorting with the previously mentioned numbers, the LLM performs the sorting without difficulty. The tradeoff is an increase in both processing time and the number of output tokens due to the additional rationale steps requiring more outputs.

We also compared the performance of various LLMs instead of GPT. Since EGoT requires the probability to evaluate the answer, we utilize the Llama 3.1 405B model and the Mixtral $8\times22$B model. The Claude 3 Haiku model does not provide the probability of the answer, therefore, we fix the probability to 1. The experiments were conducted using 10 samples for sorting 256 data. During the evaluation, both Llama and Mixtral, in contrast to GPT-4o mini, consistently assign a score of

Table 1: Results of the Number Sorting experiment (GPT-4o mini)

| 128 Elements | IO | CoT1 | CoT2 | ToT | GoT | EGoT | EGoT* |
|---|---|---|---|---|---|---|---|
| Accuracy | 90.25% | 72.13% | 90.41% | **92.28%** | 90.98% | 92.09% | 91.70% |
| Number of Errors | 14.07 | 38.79 | 13.87 | **10.87** | 11.88 | 10.94 | 11.45 |

| 256 Elements | | | | | | | |
|---|---|---|---|---|---|---|---|
| Accuracy | 70.71% | 49.50% | 83.17% | 75.58% | 84.37% | **88.31%** | 87.94% |
| Number of Errors | 119.51 | 154.57 | 49.25 | 65.74 | 40.93 | **34.54** | 35.19 |

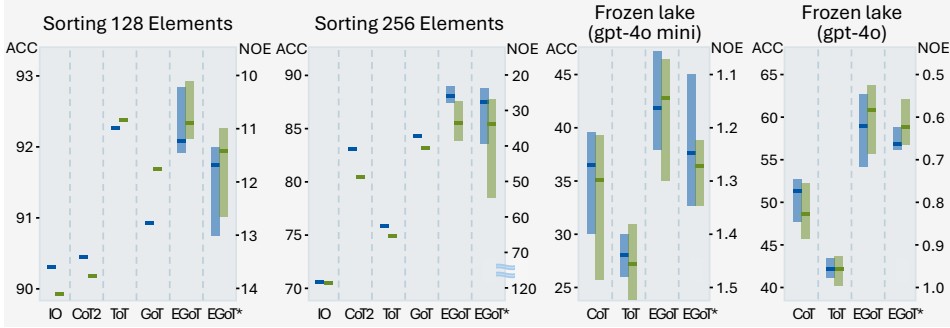

Figure 3: Figure shows Min, Max, and Average for multiple experiments. The blue line on the left of each graph represents accuracy and the green line on the right represents a number of errors. The bars represent the maximum and minimum values, and the darker color in the middle represents the average. In the sorting problem, IO, CoT2, ToT, and GoT architectures were included as comparative models, and the experiment was performed only once. The higher the ACC, the better, the lower the NOE, the better.

Table 2: Results of the 256 Number Sorting experiment using various LLMs

| Llama 3.1 405B | CoT | ToT | GoT | EGoT |
|---|---|---|---|---|
| Accuracy | 91.59% | 92.05% | 94.09% | **95.85%** |
| Number of Errors | 22.53 | 21.3 | 16.4 | **11.5** |
| | | | | |
| **Mixtral 8×22B** | | | | |
| Accuracy | 82.91% | 71.91% | 83.85% | **89.05%** |
| Number of Errors | 73.63 | 83.6 | 44.6 | **30.67** |
| | | | | |
| **Claude 3 Haiku** | | | | |
| Accuracy | 92.10% | **97.62%** | 94.38% | 95.00% |
| Number of Errors | 20.4 | **6.2** | 14.6 | 12.9 |

100 in EVALUATIONNODE. In such cases, we request the LLM for the score again. The results of these experiments are presented in Table 2.

## 5.2 FROZEN LAKE

Table 3 and Table 4 present the experimental results for the Frozen Lake problem. In the experiment, EGoT and EGoT* outperform the other architectures. To evaluate the consistent performance, five experiments were conducted using GPT-4o mini, and three experiments were conducted using GPT-4o. The results are shown in Figure 3. GoT is applicable only when the problem can be divided into sub-problems, whereas Frozen Lake cannot be broken down into smaller parts. Therefore, we cannot compare GoT in this experiment. When the rationale simply instructs the model to understand the positions of holes and tiles, the LLM often becomes confused. However, when the LLM explicitly writes the coordinates next to the input before attempting to understand the positions of the holes and tiles, its performance improves.

## 5.3 EGoT'S ADVANTAGES

EGoT shows the benefits of utilizing rationale information in prompt engineering instead of focusing only on the LLM's answer. It also proposes a structure to improve prompt engineering performance by leveraging the effect of LLM's temperature. Therefore, EGoT has two main advantages.

First, EGoT generalizes the problem by generating the prompts to enhance the basis prompt. The basis prompt contains only the rule and rationale step of the problem, and the child node enhances

Table 3: Results of the Frozen Lake experiment (GPT-4o mini)

| 5 by 5 with 8 holes | CoT | ToT | EGoT | EGoT* |
|---|---|---|---|---|
| Accuracy | 36% | 28.1% | **43%** | 41% |
| Number of Errors | 1.33 | 1.38 | **1.13** | 1.14 |

| 5 by 5 with 10 holes | | | | |
|---|---|---|---|---|
| Accuracy | 36.3% | 27.6% | **41.0%** | 34.0% |
| Number of Errors | 1.28 | 1.54 | **1.15** | 1.43 |

Table 4: Results of the Frozen Lake experiment (GPT-4o)

| 5 by 5 with 8 holes | CoT | ToT | EGoT | EGoT* |
|---|---|---|---|---|
| Accuracy | 50.8% | 39.7% | **58.8%** | 53.3% |
| Number of Errors | 0.83 | 1.03 | 0.64 | **0.62** |

| 5 by 5 with 10 holes | | | | |
|---|---|---|---|---|
| Accuracy | 51.7% | 44.4% | 59.0% | **60.3%** |
| Number of Errors | 0.80 | 0.89 | **0.55** | 0.60 |

the prompt by appending only the parent's rationale output. In all experiments, EGoT demonstrates high performance, showing that the enhancing prompt is effective.

Second, EGoT dynamically adjusts the temperature and requests a confidence score from the LLM based on both the score itself and the probability of the corresponding token. Cosine annealing is used to control the temperature, enabling the exploration of diverse answers and rationales in the early stages. Obtaining a variety of rationales helps identify issues in problem formulation and refine prompt engineering more effectively. In the end, the low temperature allows us to focus on more accurate answers rather than diversity.

### 5.4 DIFFERENCES BETWEEN EGoT AND EGoT*

EGoT* does not include dynamic temperature control, which leads to continual exploration of diverse solutions. This exploration helps maintain greater diversity in the responses. In tasks where the LLM performs well, EGoT can identify the correct answer, rather than focusing on maintaining response diversity. However, in tasks where the LLM performance is lower, exploration may lead to better answers. EGoT can sometimes exhibit lower performance when it utilizes only a portion of the provided rationale instead of considering all of it. For this reason, EGoT* performs similarly to EGoT on average when solving Frozen Lake problems. However, when solving sorting problems, EGoT demonstrates better performance.

### 5.5 DIFFERENCES WITH OTHER ARCHITECTURES

Since EGoT relies on an LLM for evaluation, it does not require external tools to verify the correctness of an answer. Mathematical problems can be easily evaluated for correctness using tools, however, general questions are more challenging to assess in this manner. EGoT does not require problem decomposition. GoT is a useful architecture if the problem can be divided hierarchically, however, it is difficult to apply to general problems where the problem cannot be partitioned. ToT functions similarly to BFS or A* in LLM-based reasoning. However, BFS or A* becomes inefficient when evaluating a large number of elements, as seen in tasks like number sorting. CoT-SC focuses solely on the answer, not the rationale, when voting for the final answer, which is efficient if the answer is a scalar. However, when the answer is a list or vector, such as experiments like number sorting or Frozen Lake, it is not as applicable as ToT. EGoT emphasizes the importance of rationale and proposes that rationale aggregation can serve a similar role to voting by continuously integrating valid rationales while discarding incorrect ones. The disadvantage of EGoT compared to the other architectures is that it requires more computational time and resources due to its larger number of nodes. Since EGoT utilizes three nodes (Answering, Evaluation, and Aggregate Rationale) to obtain

a single answer, it requires approximately three times the time and computational cost to generate the same number of answers.

# 6 RELATED WORK

## 6.1 CHAINING ARCHITECTURE AND RATIONALE STEP

There are several Prompt engineering architectures, including CoT (Wei et al., 2022), CoT-SC (Wang et al., 2023b), ToT (Long, 2023; Yao et al., 2024), EoT (Yin et al., 2023), and GoT (Besta et al., 2024). Various methods for evolving CoT and voting on the results of CoT have been proposed. Some papers emphasize the correct answer, while others emphasize the rationale. EGoT utilizes a method to construct its architecture based on EoT and Determlr (Sun et al., 2024).

CoT emphasizes the importance of rationale and CoT-SC, in contrast, focuses on the correct answer rather than the rationale. The importance of providing rationale steps in prompts is widely recognized, and this leads to research on which rationale steps should be included (Xu et al., 2024). Generally, this involves summarizing the input (Zhang et al., 2023), separating the steps, providing the feedback in the input (Yuan et al., 2024; Madaan et al., 2024), and providing an explanation of the input (Yugeswardeenoo et al., 2024). Villarreal-Haro et al. (2024) and Yin et al. (2024) demonstrate the effectiveness of two strategies: incorporating negative information into the rationale and evaluating the rationale along with its probability, both of which enhance rationale performance. These findings support the validity of the rationale step in EGoT.

## 6.2 TEMPERATURE CONTROL AND EVALUATION LLM RESPONSE

Temperature increases LLM's response diversity, and it also affects the performance of answers. Zhu et al. (2024) show the performance increase by adapting temperature with token confidence. To evaluate LLM responses, voting (Li et al., 2022; Du et al., 2024), debating (Liang et al., 2023; Xiong et al., 2023) and scoring (Lee et al., 2024a) are utilized. Since evaluating LLM response affects the performance of the architecture significantly, external tools (Gou et al., 2024) are used to evaluate the confidence level of LLM responses (Zhu et al., 2023). Motivated by these methods, we utilize debating to obtain the answer by providing the rationale of the parent node to the LLM for inferring the correct answer. Additionally, we perform self-evaluation by requesting the score for a single token from the LLM, rather than utilizing the entire set of responses, such as rationales, in the EVALUATIONNODE. We define confidence by utilizing the score and the probability of the token responded by the LLM to self-evaluate.

# 7 CONCLUSION

Prompt engineering is an area of study that is key to effectively utilizing LLMs, maximizing the advantage of LLMs: the applicability of the model to a wide variety of problems without training. Fine-tuning is essential when an LLM needs to acquire specialized skills for certain tasks, however, it can reduce generalization capabilities and tends to be costly and resource-intensive. Chain of Thought (CoT) approach enhanced the ability to reason in general situations, recently various architectures evolved methodologies that are more effective for special cases.

We emphasize that the performance of LLMs is already enough to enable automated solutions for intuitive problems. While reasoning strategies may vary based on individual needs and problem requirements, the EGoT architecture demonstrates broad applicability and consistently improves performance. Our work reemphasizes the importance of rationale, and its concise architecture suggests the possibility of prompt engineering for a wide variety of problems.

Improving the performance of the LLM is also important, obviously. We tried to compare chess puzzles to verify the performance of EGoT architecture. However, despite adding a rule in the prompts that no piece except the knight can jump, GPT-4o mini thinks it can jump over a piece in the middle of a move. As a result, no architectures can find a move that captures the opponent's piece and checkmates, and the performance is not enough to compare results. Therefore, we hope that prompt engineering techniques improve with LLM performance improvement.

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

# A APPENDIX

## A.1 EVALUATING COSTS: A COMPARISON TO GoT

Table 5: Cost Comparison for solving one problem (GPT-4o mini) in USD

| 128 Elements | Average | Minimum | Maximum |
|---|---|---|---|
| EGoT | 0.03844 | 0.02922 | 0.04404 |
| GoT | **0.03246** | **0.02815** | **0.03658** |
| **256 Elements** | | | |
| EGoT | **0.04302** | 0.03484 | 0.06855 |
| GoT | 0.05289 | **0.03353** | **0.05852** |

The number of nodes in the experiment is adjusted through iterative testing to ensure that the cost difference from GoT remains minimal. As a result, while the number of subnodes and tokens per node in EGoT is higher compared to GoT, the total number of nodes in the graph is reduced, ensuring that the overall cost difference is not substantial. For the sorting 128 elements, GoT generally exhibits approximately 20% lower cost; however, it does not hold for the sorting 256 elements. This discrepancy can be attributed to EGoT placing greater emphasis on rationale within the prompt and requesting more detailed responses. Consequently, even as the problem and correct answer length increases, the total input/output tokens increase only marginally. In contrast, GoT focuses primarily on the answer, leading to a proportional increase in input/output tokens as the problem length grows.

Mathematically, the total output tokens can be represented as $A + R$, where A denotes the number of answer tokens and $R$ represents the number of rationale tokens. EGoT generally has $R > A$, so even with an increase in $A$, the rate of increase in the total token count $A + R$ is relatively smaller. On the other hand, GoT typically has $R < A$, meaning that as $A$ increases, the rate of increase in the total token count is greater. For sorting 256 elements, EGoT has a lower average cost, while GoT has lower costs in both minimum and maximum. When the LLM solves difficult problems, it tends to generate uncertain or additional responses, which increases the total number of tokens. Conversely, for simpler problems, a more concise answer is produced, resulting in fewer tokens.

The reason for the cost difference in maximum is that EGoT has conditional statements. EGoT queries again for uncertain answers to increase the confidence of the answer at the ANSWERINGN-ODE and EVALUATIONNODE. If this process occurs at every node, it would require a higher cost. There is a higher cost increase for problems that are difficult for LLM to solve. However, in general, EGoT performs better than GoT even though its cost is lower.

## A.2 METHOD NODE PROMPT EXAMPLE (SORTING PROBLEM)

| **Human Message** |
|---|
| <Goal> |
| - When a list of numbers is entered, answer the question about how to sort them. |
| - Answer the questions about how to evaluate whether the sorted numbers found by this method is correct. |
| </Goal> |
| |
| <Instruction> |
| - Answers the method based on keywords. |
| - The evaluation methods should be done by LLM, so only print heuristic answers. |
| - Answer without being abstract, only considering the method with the best performance. |
| </Instruction> |
| **LLM Response** |
| Method: |
| - QuickSort |
| Evaluation Method: |
| - Check if the sorted list is in ascending order by comparing each element with the next one. |
| - Count the number of inversions in the original list and the sorted list; the number of inversions should decrease. |
| - Verify that the sorted list contains the same elements as the original list by checking for duplicates and ensuring all elements are present. |

Table 6: METHODNODE Prompt Example.

Table 6 is an example of a real prompt we used. The method and evaluation method may not be effective, however, we used it to minimize human intervention.

A.3 ANSWERING NODE PROMPT EXAMPLE (SORTING PROBLEM)

| **Human Message** |
|---|
| . . . 
 [Provide the method and data to resolve the problem.] 
 . . . 

 <Rationale_Step> 
 - Rationalize through 4 processes and must provide a step-by-step explanation in detail. 
 - First, accurately determine and write the all numbers in Input_Data. 
 - Be sure to sort the numbers and write down the rationale in detail using the information provided in the previous step. 
 - Make sure you explain why the sorting is correct with a rationale for the entire process. 
 - Finally, answer the process of deducing the correct answer by synthesizing the previous process. 
 </Rationale_Step> 

 . . . 
 [Provide the information from the upper nodes.] 
 . . . 

 {% if aggregate_rationale %} 
 <Hint> 
 - Refer to the hint, but they are not always accurate, so verify the hint by checking the input data and using them. 
 {% for idx in range(aggregate_rationale\|length) %} 
 - Hint_{{ idx }}: {aggregate_rationale[{{ idx }}]}{% endfor %} 
 </Hint> 
 {% endif %} |
| **LLM Response** |
| Answer: 
 - 6, 20, 20, 25, . . . 
 Answer Rationale: 
 - First, I will identify all the numbers in the input list, which are: [1348, 147, 1454, 1111, . . . Next, using the QuickSort algorithm, I will sort this list in ascending order. Using QuickSort, I will choose a pivot and divide the list into smaller sub-lists of numbers less than or greater than the pivot, then further sort those sub-lists until the entire list is sorted. The resulting sorted list is: [6, 20, 20, 25 . . . |

Table 7: ANSWERINGNODE Prompt Example. (Jinja2 template)

Table 7 is the prompt that requests the LLM to answer the question. Because we emphasize the importance of rationale, we set four rationale steps and gather effective rationale. The top node does not have aggregate rationale information, while the lower nodes have aggregate rationale information received from the upper nodes. Therefore, the prompt utilizes jinja2 template and forwards this information to the lower nodes at the bottom of the prompt with the Hint tag.

## A.4 EVALUATION NODE PROMPT EXAMPLE (SORTING PROBLEM)

| **Human Message** |
|---|
| . . .
[Provide the question, the answer from ANSWERINGNODE, and request a score.]
. . .

<Evaluation_Rationale_Step>
- Rationalize through 4 processes and must provide a step-by-step explanation in detail.
- First, accurately determine and write all the numbers in Input_Data.
- Be sure to score each number included in the sorted data and write down the rationale in detail using the information provided in the previous step.
- Make sure you explain why the sorting is correct with a rationale for the entire process.
- Finally, answer the process of deducing the correct answer by synthesizing the previous process.
</Evaluation_Rationale_Step>

<Evaluate_Methods>
{% for idx in range(evaluate_methods\|length) %}
- {evaluate_methods[{{ idx }}]}{% endfor %}
</Evaluate_Methods> |
| **LLM Response** |
| Accuracy:
- 80
Evaluation Rationale:
. . . For instance, the number '680' appears in the sorted data but is not present in the input data. Additionally, the number '994' is also present in the sorted data but is missing from the input. . . . Therefore, while the majority of the numbers are sorted correctly, the presence of these discrepancies warrants a score below 90. |

Table 8: EVALUATIONNODE Prompt Example. (Jinja2 template)

Table 8 is the prompt that requests the LLM to evaluate the response. The Evaluation Rationale Step is almost identical to table 7. However, only the content of step 2 changes slightly because it needs to evaluate the score. By parsing the log probability of the number "80" in the LLM response, the confidence is calculated as 0.607.

## A.5    AGGREGATE RATIONALE NODE PROMPT EXAMPLE (SORTING PROBLEM)

| **Human Message** |
| --- |
| <Input_Rationales> |
| - Evaluation_Rationale: {evaluation_rationale} |
| - Answer_Rationale: {answer_rationale} |
| </Input_Rationales> |
| |
| <Instruction> |
| <Aggregate_Rationale> |
| - Don't print the successes or valid, just print the incorrectly missing numbers. |
| - Integrate the flaws that are problematic or invalid. |
| - Be sure to consider the previous two steps, aggregate the Input_Rationales. |
| </Aggregate_Rationale> |
| <Inaccurate_Info> |
| - Inaccurate_Info is information to recheck regarding the missing numbers, as it seems to be inaccurate. |
| - If the number information in two rationales is different, the number should be included in the inaccurate information. |
| </Inaccurate_Info> |
| </Instruction> |
| **LLM Response** |
| Aggregate Rationale: |
| - The sorted list contains discrepancies, including numbers that are not present in the original input data, such as '680' and '994'. … |
| Inaccurate Info: |
| - The numbers '680' and '994' are present in the sorted data but are missing from the input list. Additionally, the number '1496' appears twice in the input list but is only represented once in the sorted list, indicating a potential error in the sorting process. |

Table 9: AGGREGATERATIONALENODE Prompt Example. (Jinja2 template)

Table 9 is the prompt that summarizes the rationale generated by the two nodes ANSWERINGNODE and EVALUATIONNODE. AGGREGATERATIONALENODE provides the rationale information by summarizing the key points. Additionally, it extracts negative information and propagates this to the lower nodes. In table 8, LLM informs that 680 and 994 are present in the input, however, the sorted result doesn't contain these numbers, therefore table 9 aggregates this information. Misinformation like 1496 also propagates, though the misinformation gradually vanishes as the graph progresses.

