# OpenReview forum: "Enhancing Graph Of Thought: Enhancing Prompts with LLM Rationales and Dynamic Temperature Control"
_ICLR.cc/2025/Conference — ICLR 2025 Poster_

### Official Review · Reviewer_fxo8 · 2024-11-02

**Soundness:** 3
**Presentation:** 3
**Contribution:** 3
**Rating:** 8
**Confidence:** 4

**Summary:**

This paper introduces a dynamic an approach of enhancing prompts with dynamic temperature control. The steps are organized in a tree-like structure, where there is an answering, an evaluation and an aggregate rationale component in each node and each node passes the information to the child nodes. The authors evaluate this approach evaluate a range of closed and open LLMs on tasks and compare with other relevant methods from the literature.

**Strengths:**

- The authors borrow ideas from other methods from the literature and combine them in a good way
- They use a dynamic temperature which initially allows for more exploration when the model has less confidence and decreases as the model gains more confidence of its answers
- They show that this method is improving on some common benchmarks using GPT4o/GPT4o mini
- This is a method that doesn't require fine-tuning which can be resource intensive and time consuming
- The paper is well-written and comprehensible

**Weaknesses:**

- This method is only evaluated with GPT4o. It would be important to see how other models are performing like Llama or Mistral.
- I think for this method the bottleneck is the model's self-evaluation and answering. For self-evaluation If it assigning incorrect scores and rationales, then the performance will also be bad no matter what method you use. In such a case it may even yield in worse performance than some of the methods you presented. For answering, you already provided the chess example. The best way around it is to fine-tune with well curated examples.
- For the most part these ideas are not novel. We've seen these ideas in papers like ToT,  $T^{2}oT$, GoT, SC, Self-Evaluation

**Questions:**

- From your benchmarking, I see that by fixing the temperature (EGoT*) the accuracy is sometimes very similar to dynamic temperature (EGoT). Would that indicate that for those instances dynamic temperature has no effect?
- Is there an established set of benchmarks for evaluating these methods? For example in the GoT paper I see that they also evaluate on sorting and document and merging tasks as you do, but they also evaluate on keyword counting and document merging and they don't evaluate on the frozen lake
- Minor mistakes:
Use “Prompt engineering” instead of “Prompted engineering”
"As a result, no architectures cannot find a move that captures the opponent’s piece". Do you mean $\textbf{"can find"}$?
- In your figure 2 it's hard for me to follow the coordinates. Would it be possible to make each cell look more distinct and add a x,y axis ticks?

---

> ### Author Response · Authors · 2024-11-18
> **Response to Reviewer fxo8 (1/3)**
>
> > **Weakness1.** This method is only evaluated with GPT4o. It would be important to see how other models are performing like Llama or Mistral.
>
> Thank you for your valuable suggestion. **We ran experiments with Llama 3.1 405B and Mistral 8 * 22B via Fireworks.**
>
> - Llama 3.1 405B
>
> | structure | EGoT (ours) | GoT | ToT | CoT |
> | --- | --- | --- | --- | --- |
> | Accuracy | **95.85%** | 94.09% | 92.05% | 91.59% |
> | Number of Errors | **11.5** | 16.4 | 21.3 | 22.53 |
>
> - Mistral 8 * 22B
>
> | structure | EGoT (ours) | GoT | ToT | CoT |
> | --- | --- | --- | --- | --- |
> | Accuracy | **89.05%** | 83.85% | 71.91% | 82.91% |
> | Number of Errors | **30.67** | 44.6 | 83.6 | 73.63 |
>
> - Anthropic claude-3-haiku
>
> | structure | EGoT (ours) | GoT | ToT | CoT |
> | --- | --- | --- | --- | --- |
> | Accuracy | 95.00% | 94.38% | **97.62%** | 92.10% |
> | Number of Errors | 12.9 | 14.6 | **6.2** | 20.4 |
>
> We used the fireworks(https://fireworks.ai/) platform to conduct experiments for open-sourced LLMs. Also, we experimented with Anthropic’s models, a commercial LLM. However, Anthropic does not provide logprob functionality, so we fixed logprob to 1. Kindly note that the other two models (Mistral, Llama) provide the logprob of their answers.
> We ran the experiment with a total of 10 sorting data and the results are shown in the table. During the experiment, Llama and Mistral, unlike GPT-4o mini, almost always answered score as 100 in the Evaluation Node, in which case we asked score one more time regardless of logprob. The results show that our idea EGoT was the most effective on both Llama and Mistral, and it is slightly effective on Anthropic without logprob.
>
> ---
>
> > **Weakness2.**
>
> Thank you very much for your thoughtful comments. Kindly note that the objectives of fine-tuning and prompt engineering are orthogonal: fine-tuning involves additional training to improve performance on specific tasks, while **prompt engineering utilizes an LLM’s existing capabilities to address a wide range of challenges.** We agree that fine-tuning is essential when an LLM needs to acquire specialized skills for certain tasks; however, it can reduce generalization capabilities and tends to be costly and resource-intensive. **In this paper, our goal is to maximize LLM effectiveness without relying on fine-tuning.** We will clarify this point in the manuscript.

---

> ### Author Response · Authors · 2024-11-18
> **Response to Reviewer fxo8 (2/3)**
>
> > **Weakness3.** For the most part these ideas are not novel. We've seen these ideas in papers like ToT, T2oT, GoT, SC, Self-Evaluation
>
> We acknowledge that some concepts in our work are inspired by existing research. However, **the key novelty of our approach lies in addressing potential challenges that arise during the integration of multiple components, enabling us to optimize the trade-off and achieve significant performance improvements.**
>
> For instance, [1] demonstrated that during token decoding, confident tokens reduced the temperature to minimize noise, while difficult tokens increased the temperature to allow the LLM to explore a variety of choices—an approach shown to be effective experimentally. Our method extends the classical exploration-exploitation framework in graph structures, applying it to prompt engineering as outlined in [1].
>
> The key innovation of our method lies in extending dynamic temperature control from individual tokens to entire prompts, while also utilizing rationales instead of directly propagating LLM-generated answers. Through experiments, we observed that for more complex tasks, LLMs often repeat incorrect answers rather than providing accurate ones. Motivated by this, we focused on propagating rationales, which led to non-trivial model performance improvement compared to baseline methods.
>
> Furthermore, LLMs often provide only positive rationales when answering open-ended questions. For instance, when asked to sort {1, 5, 4, 3}, LLMs typically do not point out that the number 2 is missing from the list, nor do they mention the omission of 4 when providing the answer {1, 3, 5}. To address this limitation, we introduced self-evaluation, incorporating both positive and negative rationales to improve answer accuracy. A related paper presented at CLEF 2024 [2] also explores the use of both positive and negative rationales, further validating the effectiveness of our approach.
>
> The table below summarizes the differences between EGoT and other existing baselines.
>
> | structure | **EGoT (ours)** | GoT | T2oT| ToT | CoT-SC | Self Evaluation |
> | --- | --- | --- | --- | --- | --- | --- |
> | Dynamic Temperature Control | O | X | O | X | X | X |
> | Automate scoring without Oracle | O | △ | X | X | O | O |
> | Information pass to the following nodes | Rationale for the answer and score | previous answer | previous answer | previous answer | voting result of previous answer | sample n answers |
> | Applicable problem diversity | Any problem is possible | Possible if the problem can be divided into subproblems | Any problem is possible | Any problem is possible | Any problem is possible | Any problem is possible |
>
> [1] Yuqi Zhu, Jia Li, Ge Li, YunFei Zhao, Zhi Jin, and Hong Mei. Hot or cold? adaptive temperature sampling for code generation with large language models. In Proceedings of the AAAI Conference on Artificial Intelligence, volume 38, pp. 437–445, 2024.
>
> [2] Kapioma Villarreal-Haro, Fernando Sanchez-Vega, Alejandro Rosales-P ´ erez, and Adri ´ an Pastor ´ Lopez-Monroy. Stacked reflective reasoning in large neural language models. ´ Working Notes of CLEF, 2024.
>
> ---

---

> ### Author Response · Authors · 2024-11-18
> **Response to Reviewer fxo8 (3/3)**
>
> > **Question1.** From your benchmarking, I see that by fixing the temperature (EGoT*) the accuracy is sometimes very similar to dynamic temperature (EGoT). Would that indicate that for those instances dynamic temperature has no effect?
>
> Kindly note that EGoT* has a higher temperature than EGoT, which **helps maintain greater diversity in the responses.** In tasks where LLMs perform well (e.g., sorting), EGoT outperformed, as the priority is finding the correct answer rather than maintaining response diversity. However, in tasks where LLM performance is lower (e.g., Frozen Lake), exploration may lead to better answers, which could explain why the overall average performance appears similar. However, **our proposed structures that effectively propagate rationale shows non-trivial performance improvement across all tasks.**
>
> ---
>
> > **Question2.** Is there an established set of benchmarks for evaluating these methods? For example in the GoT paper I see that they also evaluate on sorting and document and merging tasks as you do, but they also evaluate on keyword counting and document merging and they don't evaluate on the frozen lake
>
> Currently, there is no well-established benchmark for this field, which is why **the datasets have evolved from the CoT paper to the GoT paper.** For instance, ToT experimented with 24 games, Creative Writing, and 5x5 Crosswords, while GoT did not include these. It is important to note that our approach is designed to address more complex problems effectively. We chose Frozen Lake over keyword counting because it is a widely recognized benchmark in reinforcement learning and presents a more challenging problem that previous works have not explored. As LLMs continue to improve, we believe they will be capable of solving more complex real-world datasets, such as those used in the **AIXCC competition organized by US DARPA for code patching.**
>
> ---
>
> > **Question3.** Minor mistakes: Use “Prompt engineering” instead of “Prompted engineering” "As a result, no architectures cannot find a move that captures the opponent’s piece". Do you mean can find?
>
> **Thank you for pointing out this error. We committed to enhancing the quality of our writing.**
>
> ---
>
> > **Question 4.** In your figure 2 it's hard for me to follow the coordinates. Would it be possible to make each cell look more distinct and add a x,y axis ticks?
>
> **Thank you for your helpful comment. We agree that this is an important clarification for understanding the paper.**
>
> For each tile in Figure 2, we labeled the numbers 0, 1, 2, 3, and 4 along the top and left sides.

---

> ### Author Response · Authors · 2024-11-25
> **Official Comment by Authors**
>
> Dear Reviewer fxo8,
>
> I hope this message finds you well. As the end of the discussion period is approaching, we have not yet received any feedback. We have made significant efforts to address the concerns you raised during the initial review, and we would be grateful for any further questions or feedback you may have. Additionally, we would be grateful if you could kindly share whether our revision has impacted your evaluation. Thank you very much for your time and valuable insights. We look forward to hearing from you soon.
>
> Sincerely,
>
> Authors of Submission6264

---

> > ### Comment · Reviewer_fxo8 · 2024-11-25
> >
> > Dear authors,
> >
> > Thanks for your clarifications. Please write the key novelty that you have stated in your paper and make it apparent. Thanks for running experiments with additional models. Now I can see more clearly the non-trivial improvement. Please include the results in the paper. This has clarified my questions and I have a more positive outlook towards the paper. I will update my rating

---

> > > ### Author Response · Authors · 2024-11-25
> > > **Official Comment by Authors**
> > >
> > > Dear Reviewer fxo8,
> > >
> > > Thank you for your valuable comments. In accordance with your suggestion, we have made the following revisions to the paper.
> > >
> > > We have introduced a novelty contribution in **line 65 of Section 1**, added additional experimental results in **Section 5.1 on line 372** (including Table 2), and discussed the differences between EGoT and EGoT* in **Section 5.4 on line 454**.
> > >
> > > Sincerely,
> > >
> > > Authors of Submission6264

---

### Official Review · Reviewer_JcuQ · 2024-11-03

**Soundness:** 2
**Presentation:** 1
**Contribution:** 2
**Rating:** 5
**Confidence:** 4

**Summary:**

This paper proposes Enhancing Graph of Thoughts (EGOT) to improve the performance of LLM on reasoning tasks. They design a flow within each node to answer the question with rationales, evaluate the answer with rationales, and consolidate the answer and evaluation rationales as inputs to the child nodes. They also introduce temperature control with cosine annealing to explore answer space at earlier nodes and refine the promising answers at later nodes. They report that the proposed methodology shows better performance on number sorting and frozen lake tasks compared to baselines including CoT, ToT, and GoT.

**Strengths:**

1. This paper goes beyond the evaluation scoring used in ToT and GoT for pruning and ranking; instead, they also leverage LLMs to provide rationales for the answers and evaluation scores to help the model refine the answers.

2. This paper introduces temperature control with cosine annealing to balance between exploration and exploitation.

3. This paper demonstrates the effectiveness of EGoT on an additional reasoning task, Frozen Lake problem, which has not been explored in any of baselines including CoT, ToT, and GoT.

**Weaknesses:**

1. The paper formulates the methodology with a graph architecture, but it does not hold certain characteristics of a graph similar to Graph of Thoughts (GoT). First, figure 1 starts with Method Node as root and section 2.2 says “heuristic methods are requested from LLM and utilized”, but it’s not clearly illustrated in the example use case (section 3.3), hence making the interaction between Method Node and other nodes confusing. Second, based on the section 3.3 example use case, the methodology starts with 3 nodes and there are no interaction between child nodes either from the same parent or different parents until the very end. Given these observations, it seems like 3 Tree of Thoughts with aggregation at the last layer. It would be good to explain more clearly on this front.

2. Some settings mentioned in the paper seem arbitrary without thorough explanation. First, for the evaluation score, the authors run evaluation once more with a temperature of 1 if the probability is lower than a threshold and different scores have different thresholds. Without looking at the specific evaluation prompt, it is not clear why we select a scale of 0-100 instead of 0-10 or 0-5. Since the evaluation requires more accuracy and lower diversity, it it not clear why we set a temperature of 1 to re-run the evaluation instead of relying on the scores generated with a temperature of 0. Could you provide justifications for these choices? Are they based on any experiment results?

3. In section 4, the authors state that “for GoT, instead of evaluating the best performing node, they select the one with medium score value, with the goal of comparing only structural performance, as solving problems with evaluation metrics is not considered to be a structural advantage.” However, it may not be convincing as EGoT also takes advantage of LLM evaluation with rationales instead of structure changes only; it may not be a fair comparison to only select the medium for GoT.

**Questions:**

1. It would be good to add the prompts used for each task and each node to appendix

2. In section 3.3, the illustrated case branches each node in depth 0 into 2 child nodes, but after depth 0, each parent node only generates 1 child node. What's the criteria to determine how many child nodes to be generated.

3. In section 3.1.1, the authors state that if the child gives the same answer as the parent, a questions is only asked once more, since it cannot be determined whether it’s the correct answer. It would be good to validate if the score or confidence level produced by the evaluation node could determine if one more run is needed.

4. Curious to understand the cost compared to other baselines.

---

> ### Author Response · Authors · 2024-11-18
> **Response to Reviewer JcuQ (1/2)**
>
> > **Weaknesses 1.**
>
> We introduced the method node because we believed that proposing specific solutions to the problem could lead to performance differences depending on the approach taken, which could undermine the strength of our structure. To avoid this, we used the methods provided by the LLM to solve the problem and referred to the method node in Section 2.2. This is not the main focus of our paper, but rather a necessary tool to maintain experimental objectivity. For example, ToT uses the DFS method to solve the sorting problem, where it asks the LLM to return the smallest element in the input list and passes the list after removing that element to the next node. In GoT, the list is divided into smaller problems and the LLM is asked to answer them. However, since our focus is not on how to solve the problem, but on how to generate and propagate rationale effectively, we omitted the method node in Section 3.3. **We appreciate this advice and added an explanation related to Method Node on line 216 of page 5 of the paper.**
>
> From the perspective of propagating temperature, a tree structure makes sense, but **when considering the propagation of the rationales, a graph structure is more appropriate.** Hence, we chose to refer to it as a graph, which is a more general concept. In Figure 2, the depth 0 rationales are shared between nodes 0, 1, 2 and nodes 3, 4, 5, 6, 7, 8. The depth 0 rationale layer encompasses all the rationales generated by nodes 0, 1, and 2, with nodes 3-8 receiving all the rationale information from nodes 0-2. Thank you for your comment, and we will ensure to provide additional clarification in the figure.
>
> ---
>
> >**Weaknesses 2.**
>
> Some parts of our paper may not have been as clear as intended, which could have led to potential misinterpretation. Please note that the evaluation was conducted with the temperature set to 0 in all cases to eliminate probabilistic responses. **Thanks for this advice, we added t=0 to the formulas on page 3 of the paper and added an explanation for t.  Also, we have fixed the temperature settings in 106, 305 and 323 lines. Thanks again for your advice.**
>
> Additionally, the reason we chose a score range of 0-100 instead of 0-10 is to **better represent the scores as percentages.** To avoid any further confusion, we revised the relevant sections to provide a clearer explanation of these concepts. We appreciate this advice and added it to page 3, section 2.4, line 136.
>
> ---
>
> >**Weaknesses 3.**
>
> If an oracle does not exist, LLM relies on the order of the inputs to assess accuracy, which can lead to inaccurate scoring and answers. To address this, we combine scoring and probability to improve the results.
>
> When experimenting with sorting 256 using the GoT architecture, we achieved an **accuracy of 86.74% (GoT)** and a number of errors of 35.13 when **selecting the nodes in the top 1/3 based on accuracy.** This demonstrates that our architecture still performs better, with **EGoT achieving 88.31% accuracy** and 34.54 errors.
>
> When we ask a high-performing LLM to provide the correct answer, it sometimes achieves over 96% accuracy. However, when we request answers from LLMs in other scenarios, their accuracy can drop below 70% (as seen in CoT experiments). According to the source code (operations_graph.append_operation(operations.Generate(1, 10))) [1] provided in the GoT paper, they obtain 10 responses and select only the best one.
>
> In fact, when we provide LLM with all 10 sorted results and ask it to identify the best one, it gives an unusual response. Given the accuracy of 10 lists (91.85%, 92.80%, 89.14%, 92.75%, 93.87%, 92.11%, 94.32%, 92.75%, 91.42%, and 93.56%) along with the input list, LLM does not evaluate all 10 lists. Instead, it only checks some of them in the order they are input and incorrectly identifies the first list with 91.85% accuracy as the most accurate. This happens because LLM tends to be less accurate when comparing too much data at once. Interestingly, 91.85% is actually the 8th best performance out of the 10 lists. Therefore, in GoT, we concluded that **selecting the best list out of 10 without an oracle is equivalent to choosing the first list in a random order, which effectively aligns with the median.**
>
> [1] https://github.com/spcl/graph-of-thoughts/blob/main/examples/sorting/sorting_128.py#L744

---

> ### Author Response · Authors · 2024-11-18
> **Response to Reviewer JcuQ (2/2)**
>
> >**Questions 1.** It would be good to add the prompts used for each task and each node to appendix
>
> **Thanks for the advice, we added an appendix to the paper with the prompts of the nodes.**
>
> Also the prompts can be found in the src/prompt/human directory within the uploaded zip file. As the graph progresses, rationales are generated and appended below the initial prompt.
>
> ---
>
> >**Questions 2.** In section 3.3, the illustrated case branches each node in depth 0 into 2 child nodes, but after depth 0, each parent node only generates 1 child node. What's the criteria to determine how many child nodes to be generated.
>
> Given that the cost increases significantly as the number of child nodes grows, **we chose not to use a large number of tokens compared to GoT and set the number of child nodes to 1.** However, if the budget allows, the architecture can support a higher number of child nodes. We have carefully considered these cost aspects, and **the details are provided in the answer to question 4 below.**
>
> ---
>
> >**Questions 3.** In section 3.1.1, the authors state that if the child gives the same answer as the parent, a questions is only asked once more, since it cannot be determined whether it’s the correct answer. It would be good to validate if the score or confidence level produced by the evaluation node could determine if one more run is needed.
>
> Thank you so much for your advice. For instance, in the case of note 1, we considered whether it would be better to ask the answering node again if the evaluation node scored low, or whether it would be more effective to propagate the negative information from the evaluation node to the child nodes and ask them. **We decided to propagate the negative information** (e.g., '770 does not exist in the original list') to the lower nodes in order to obtain more accurate answers, **rather than continuing the self-loop.**
>
> ---
>
> >**Questions 4.** Curious to understand the cost compared to other baselines.
>
> In the paper, we mentioned that the cost could be nearly three times higher. However, in Figure 2, we reduced the number of nodes compared to other baselines, so the actual cost is **approximately 0.8 to 1.2 times higher.** Specifically, in our experiment, the output token cost was higher due to the detailed consideration of the rationale, and the input token cost was also elevated because we attached the rationale to the input. To balance performance and cost, we experimentally reduced the depth of the nodes and the starting node, achieving performance that exceeds the baselines while maintaining a cost similar to that of GoT on average. Below are the results comparing our experiment with GoT, and in some cases, we observe that our experiment uses a lower price.
>
> - sorting 128 (EGoT costs 18.42% more than GoT.)
>
> | structure | average | min | max |
> | --- | --- | --- | --- |
> | EGoT (ours) | 0.03844$ | 0.02922$ | 0.04404$ |
> | GoT | 0.03246$ | 0.02815$ | 0.03658$ |
>
> (0.03844/ 0.03246 = 1.1842)
>
> - sorting 256 (EGoT costs 18.67% less than GoT.)
>
> | structure | average | min | max |
> | --- | --- | --- | --- |
> | EGoT (ours) | 0.04302$ | 0.03484$ | 0.06855$ |
> | GoT | 0.05289$ | 0.03353$ | 0.05852$ |
>
> (0.04302 / 0.05289 = 0.8133)

---

> ### Author Response · Authors · 2024-11-25
> **Official Comment by Authors**
>
> Dear Reviewer JcuQ,
>
> I hope this message finds you well. As the end of the discussion period is approaching, we have not yet received any feedback. We have made significant efforts to address the concerns you raised during the initial review, and we would be grateful for any further questions or feedback you may have. Additionally, we would be grateful if you could kindly share whether our revision has impacted your evaluation. Thank you very much for your time and valuable insights. We look forward to hearing from you soon.
>
> Sincerely,
>
> Authors of Submission6264

---

> ### Author Response · Authors · 2024-12-01
> **Official Comment by Authors**
>
> Dear Reviewer JcuQ,
>
> I hope this message finds you well. As the end of the extended discussion period is approaching, we have not yet received any feedback. We have made significant efforts to address the concerns you raised during the initial review, and we would be grateful for any further questions or feedback you may have. Additionally, we would be grateful if you could kindly share whether our revision has impacted your evaluation. Thank you very much for your time and valuable insights. We look forward to hearing from you soon.
>
> We have made the following revisions to the paper in response to the concerns raised by Reviewer JcuQ. For a detailed overview of the changes, we kindly invite you to refer to our official comments.
>
> **W1**: We have added an explanation regarding the Method Node on lines 87-88 (p.2) and line 216 (p.5).
>
> **W2**: Revisions have been made to lines 116, 305, and 321.
>
> **Q1**: An appendix containing the prompts of the nodes has been added.
>
> **Q2, Q4**: We have added Appendix A.1.
>
> Sincerely,
>
> Authors of Submission6264

---

> > ### Comment · Reviewer_JcuQ · 2024-12-02
> >
> > Thanks for the detailed response! I got clarification for weakness 1/2 and all questions. For weakness 1, it would be good to make figure 1 clearer to show that the rationales are shared for all nodes at the next depth.
> >
> > However, I still have two concerns. First, I think the info provided for weakness 3 is an interesting finding, but I'm not sure if an LLM will always identify the first one as the best one based on my experience. It would be good to completely follow the GoT pipeline to provide baseline numbers, and share some quantitive numbers to support the finding. Second, according to the response to question4 and line 278, it may require some experiments on data with ground truth to determine the optimal number of layers based on quality and/or cost, for each task. It would be good to have a mechanism to automatically determine when to stop and show costs for additional tasks, especially for tasks like document merging which requires heavy text rationales. Given these two concerns, I'll keep my current score.

---

> > > ### Author Response · Authors · 2024-12-03
> > > **Official Comment by Authors**
> > >
> > > Dear Reviewer JcuQ,
> > >
> > > We sincerely appreciate your valuable feedback and the clarifications provided.
> > >
> > > I understand that depicting the tree-like structure in Figure 1 may lead to some confusion. As it is not possible to revise the paper at this time, I will add more nodes to the figure in the future to clarify that the rationale is shared at the same depth.
> > >
> > > I also fully understand your continued concerns regarding the two issues, and additional explanations have been included to address them.
> > >
> > > **We would greatly appreciate it if you could once again consider that EGoT does not involve an oracle.**
> > >
> > > ### **First**
> > >
> > > There are two types of oracles:
> > >
> > > 1. Oracle that determines the closest answer to the correct one
> > >
> > > - This oracle selects the most correct answer. It can be represented in code as follows:
> > >
> > > - best_answer_accuracy = max(best_answer_accuracy, ORACLE(current_answer))
> > >
> > > - In this case, accuracy tends to increase with respect to time and token count.
> > >
> > > 2. Oracle that determines whether an answer is correct or not
> > >
> > > - This oracle simply evaluates whether the answer is correct.
> > >
> > > - As a result, previous responses have little impact on subsequent answers, and the accuracy tends to fluctuate.
> > >
> > > As shown in Table 1 of the paper, for sorting 256, ToT achieved 75.58%, while CoT2 achieved 83.17%. If the LLM is possible to find an answer close to the correct one, similar to Oracle 1, the performance of ToT [1] should always be higher than that of CoT2. However, ToT often performs worse than CoT2, which suggests that **the LLM struggles with evaluation when both the problem and the answer are given.** Although there are cases where the LLM makes correct judgments, those instances are fewer than expected. Therefore, we set the baseline at half and conducted the experiment accordingly.
> > >
> > > When Oracle 1 is available, a brute force approach that requests multiple responses will reach the upper limit of the model's performance. In other words, regardless of the structure or approach, it will converge toward the maximum performance that the LLM can achieve. The advantage of GoT, as it can reach faster than brute force for problems that can be decomposed.
> > >
> > > On the other hand, the situation we aimed to compare was the case without an oracle. Our goal was to compare methods for extracting the model's performance in the absence of an oracle.
> > >
> > > ### **Second**
> > >
> > > I understand this question as being similar to the concept of early stopping. In our implementation, you can see in line 91 of the src/graph/sorting_graph.py file that we attempted a conditional edge in the graph. As you mentioned, the intention was to automatically terminate the graph and output the final answer when a specific score or token count is reached. **However, in the absence of an oracle, it is difficult to accurately evaluate the progression of the structure, so we did not pursue this idea.** Predicting the number of output tokens generated by the LLM in advance is challenging, as the number of tokens depends on the difficulty of the problem and the prompt content. Therefore, calculating this in advance is nearly impossible. As you suggested,  in order to determine the size of a single graph, it is necessary to first conducted it through an initial comprehensive experiment.
> > >
> > > For dynamic temperature adjustment, we used cosine annealing. This method involves adjusting the temperature passed to the sub-nodes based on the number of nodes. Thus, rather than setting up a very large graph and terminating it midway, we judged that completing a single graph and letting it run to completion is a better approach. Regarding early stopping, it is typically performed by setting a min_delta, which is usually determined experimentally. **Since early stopping is also based on empirical observation, we would appreciate it if you could kindly reconsider this.**
> > >
> > > **Considering these points, we kindly request that you reconsider your evaluation.**
> > >
> > > Sincerely,
> > >
> > > Authors of Submission6264
> > >
> > > [1] https://github.com/spcl/graph-of-thoughts/blob/main/examples/sorting/sorting_128.py#L118

---

### Official Review · Reviewer_X27s · 2024-11-03

**Soundness:** 3
**Presentation:** 2
**Contribution:** 3
**Rating:** 6
**Confidence:** 3

**Summary:**

The paper present an architecture for prompt engineering based on Graph of Thoughts, aimed at supporting LLM reasoning for complex reasoning tasks. It is evaluated on the frozen lake problem.

**Strengths:**

Addressing advanced methods for prompting LLMs is valuable.
The proposed method starts from a reasonable basis.

**Weaknesses:**

The paper should be enhanced to resolve readability issues and exemplification of the steps in their workflow, e.g., the nested steps are not explained following a clear workflow, and are not associated with the claimed approach for dynamic temperature control. I acknowledge that the authors provided some context to understand their results, and took several steps to improve the narrative of the paper.

**Questions:**

Can you provide a clear workflow and an example of a concrete prompt on a real world reasoning problem?
Can you formalise the steps used in the method?
Can you experiment with the method on existing LLMs?
I acknowledge that the authors made efforts to clarify the workflow and broadened the set of tested LLMs.

---

> ### Author Response · Authors · 2024-11-18
> **Response to Reviewer X27s (1/2)**
>
> > **Weaknesses 1.** The paper is hardly readable, with non-sequiturs, very generic steps and formulas that are not exemplified.
>
> **Our research shows the benefits of utilizing rationale information in prompt engineering instead of focusing only on LLM's answer, and proposes a structure to improve prompt engineering performance by utilizing the effect of LLM's temperature.**
>
> LLM utilizes probability and temperature to generate a variety of answers. However, if the answers converge to a certain point, the influence of noise in the process should be reduced and the answer should be generated accurately. The existence of an external oracle that provides the correct answer can guide the reasoning process accurately, but in general, creating an oracle is not an easy problem.
>
> That's why we proposed an approach to generate the answer without an ORACLE.
>
> Since there is no oracle, we needed to evaluate the performance of the answers and utilized evaluation nodes. To make the answers converge to one, we performed dynamic temperature control.
>
> As the reasoning process progressed, we thought about what information to pass between the nodes, and we experimented and found that passing the rationale was more effective than passing the answer, so we proposed an aggregate rationale node to pass the rationale.
>
> If you still have any remaining concerns regarding unclear or illogical aspects of the paper, we would be more than happy to discuss them in detail during the discussion stage.
>
> ---
>
> > **Weaknesses 2.** The evaluation is performed on a toy example and it is not easy to figure out how to address a real world complex reasoning task.
>
> Thank you for your comment. **Kindly note that we have followed the experimental setup convention established by previous works in this field.** For instance, the ToT paper also conducted experiments using the 24-game example [1], the GoT paper conducted experiments using the sorting example [2]. Furthermore, we present a more challenging problem, such as the Frozen Lake problem, which has not been explored in previous works. Additionally, we acknowledge that the current knowledge of LLMs is insufficient to effectively address real-world problems. For example, the AIXCC competition held by DARPA in September is a highly funded event, where one of the teams that advanced to the finals identified bugs in only two categories among numerous challenges, earning a $2 million prize. If you could provide examples of real-world problems addressed by other related papers, we would be happy to compare the performance of our model with other baselines.
>
> ---
>
> > **Weakness 3.** The nested steps are not explained following a clear workflow, and are not associated with the claimed approach for dynamic temperature control.
>
> **Thank you for your feedback. If you could kindly specify which parts you found unclear or irrelevant, I would be happy to revise the paper or provide further clarification during the discussion stage.**

---

> ### Author Response · Authors · 2024-11-18
> **Response to Reviewer X27s (2/2)**
>
> > **Question 1.** Can you provide a clear workflow and an example of a concrete prompt on a real world reasoning problem?
>
> **Thank you for seeking clarification on this matter. In summary, the overall workflow in this paper operates in three stages: (Stage 1) obtaining a answer to resolve the problem from the LLM, (Stage 2) asking the LLM to evaluate the response and assign a score, and (Stage 3) collecting the LLM’s rationale from Steps 1 and 2 and forwarding it to the next node.**
>
> Our proposed method contributes to Stage 3 by applying a technique that enhances the architecture's performance. Specifically, we adjust the temperature as the process progresses, thereby increasing the confidence in the answers. We would be happy to modify a flow diagram illustrating (figure 2) this process in the revised paper.
>
> >**Question 2.** Can you provide an example of a concrete prompt on a real world reasoning problem?
>
> **Kindly note that we have followed the experimental setup convention established by previous works in this field.** The Frozen Lake problem, which we have provided in the paper, is a commonly used problem in real-world scenarios. For example, if a road is blocked and we need to find a detour, we can use road information to navigate the way. Kindly note that, due to the significance of Frozen Lake problems in real-world scenarios, it is widely recognized as a benchmark in reinforcement learning. **The prompts used in our experiments are included as an appendix at the end of the paper.**
>
> >**Question 3.** Can you formalize the steps used in the method?
>
> We will explain this process again using Figure 1 from the paper. In Figure 1, we begin with the 'method' node, which outlines how the LLM can solve the problem. The structure then moves to the 'answering' node, where the problem and solution approach are posed to the LLM to generate an initial response (Stage 1). This answer is likely to be incorrect due to a high temperature setting. Next, we pass the response from Stage 1 to the 'evaluation' node, asking the LLM to assess its accuracy and identify any errors (Stage 2). Then, we use the 'aggregate rationale' node to summarize and combine the LLM's responses (Stage 3).
>
> After performing these three stages, we ask the LLM to answer again (Stage 1), providing it with the rationale information gathered previously. Since this aggregate rationale could potentially confuse the LLM, following this guidance helps reduce errors and improve the answer's accuracy. By iterating this process like a graph and propagating the rationale information, the system gradually converges toward the correct answer, which is achieved by lowering the temperature. A detailed explanation of the nodes' functions and temperature control is provided in the paper. **Please reach out again if any part is unclear.**
>
> >**Question 4.** Can you experiment with the method on existing LLMs?
>
> **We conducted experiments using the most widely used LLMs, GPT-4o and GPT-4o Mini, as outlined in our paper.** Additionally, we ran experiments with Llama 3.1 405B and Mistral 8 * 22B via Fireworks.
>
> - Llama 3.1 405B
>
> | structure | EGoT (ours) | GoT | ToT | CoT |
> | --- | --- | --- | --- | --- |
> | Accuracy | **95.85%** | 94.09% | 92.05% | 91.59% |
> | Number of Errors | **11.5** | 16.4 | 21.3 | 22.53 |
>
> - Mistral 8 * 22B
>
> | structure | EGoT (ours) | GoT | ToT | CoT |
> | --- | --- | --- | --- | --- |
> | Accuracy | **89.05%** | 83.85% | 71.91% | 82.91% |
> | Number of Errors | **30.67** | 44.6 | 83.6 | 73.63 |
>
> - Anthropic claude-3-haiku
>
> | structure | EGoT (ours) | GoT | ToT | CoT |
> | --- | --- | --- | --- | --- |
> | Accuracy | 95.00% | 94.38% | **97.62%** | 92.10% |
> | Number of Errors | 12.9 | 14.6 | **6.2** | 20.4 |
>
> We used the fireworks(https://fireworks.ai/) platform to conduct experiments for open-sourced LLMs. Also, we experimented with Anthropic’s models, a commercial LLM. However, Anthropic does not provide logprob functionality, so we fixed logprob to 1. Kindly note that the other two models (Mistral, Llama) provide the logprob of their answers.
> We ran the experiment with a total of 10 sorting data and the results are shown in the table. During the experiment, Llama and Mistral, unlike GPT-4o mini, almost always answered score as 100 in the Evaluation Node, in which case we asked score one more time regardless of logprob. The results show that our idea EGoT was the most effective on both Llama and Mistral, and it is slightly effective on Anthropic without logprob.

---

> ### Author Response · Authors · 2024-11-25
> **Official Comment by Authors**
>
> Dear Reviewer X27s,
>
> I hope this message finds you well. As the end of the discussion period is approaching, we have not yet received any feedback. We have made significant efforts to address the concerns you raised during the initial review, and we would be grateful for any further questions or feedback you may have. Additionally, we would be grateful if you could kindly share whether our revision has impacted your evaluation.
> Thank you very much for your time and valuable insights. We look forward to hearing from you soon.
>
> Sincerely,
>
> Authors of Submission6264

---

> > ### Comment · Reviewer_X27s · 2024-11-26
> >
> > I thank for the clarifications provided. My doubts were indeed due to narrative and exemplification. I'll raise my scores consequentially. Please take care of double-checking the readability of the paper.

---

> > > ### Author Response · Authors · 2024-11-27
> > > **Official Comment by Authors**
> > >
> > > Dear Reviewer X27s,
> > >
> > > We greatly appreciate the time you have taken to review our research. We hope that our comments have conveyed how challenging it is to solve real-world problems using only LLMs. Thank you for your understanding, and we will continue to work on improving the readability of the paper. If our submission is still considered to be marginally below the acceptance threshold, we would greatly appreciate any additional feedback on unresolved issues, as it would be extremely helpful for the further development of our research. We sincerely value your feedback.
> > >
> > > Sincerely,
> > >
> > > Authors of Submission6264

---

> ### Author Response · Authors · 2024-12-01
> **Official Comment by Authors**
>
> Dear Reviewer X27s,
>
> We sincerely appreciate your valuable feedback and the clarifications provided. In response to your suggestion, we have made several efforts to enhance the readability of the paper. We believe these revisions will contribute to a more accessible and cohesive presentation of the content.
>
> We have made the following revisions to the paper in response to the concerns raised by Reviewer X27s. For a detailed overview of the changes, we kindly invite you to refer to our official comments.
>
> **W1**: We have modified Section 1 (lines 65-66) and Section 5.3 (lines 426-431).
>
> **Q1, Q3**: We have modified Section 2.1 (lines 83-88).
>
> **Q4**: We have added content from lines 372 to 377 in Section 5.1 and Table 2.
>
> We believe that we have addressed nearly all of the concerns, but we would greatly appreciate your insight into why the evaluation remains marginally below acceptance. If there are any additional areas for improvement or concerns you may have, we would greatly appreciate your feedback.
>
> Sincerely,
>
> Authors of Submission6264

---

### Official Review · Reviewer_Gedu · 2024-11-05

**Soundness:** 3
**Presentation:** 3
**Contribution:** 2
**Rating:** 6
**Confidence:** 2

**Summary:**

This paper introduces EGoT, an architecture for enhancing large language model (LLM) performance by dynamically generating prompts and answers using a base prompt. It utilizes cosine annealing for temperature control to refine answer confidence and log probabilities to improve accuracy without needing additional training or examples. EGoT demonstrates consistent, high-quality output across tasks by continuously appending rationale steps and focusing on diverse yet accurate responses.

**Strengths:**

- The use of rationale-driven approaches is well-motivated, and experiments show EGoT achieving higher performance in tasks like number sorting and frozen lake.
- Example use cases effectively illustrate the mechanism’s function, though additional examples of negative rationales would enhance understanding.

**Weaknesses:**

- It would be beneficial to include cases where EGoT fails and scenarios where the gap between EGoT and EGoT* is more pronounced.
- Experiments were limited to GPT-4o models. Findings might differ from other model types.
- How the aggregate rationale node functions is unclear. How effectively can LLMs aggregate rationales?

**Questions:**

- Given that EGoT requires three times the resources compared to other mechanisms, for which tasks does this approach offer an advantage, and in which cases might it be less suitable?
- Is there any rationale behind selecting these specific experimental setups—document merging, number sorting, and frozen lake?

---

> ### Author Response · Authors · 2024-11-18
> **Response to Reviewer Gedu (1/3)**
>
> > **Weaknesses 1.** It would be beneficial to include cases where EGoT fails and scenarios where the gap between EGoT and EGoT* is more pronounced.
>
> > **Weaknesses 1.1.** EGoT fail case
>
>  Thank you for the suggestion. As mentioned in the paper, we conducted five trials with different random seeds. The results presented in this answer illustrate the fail case by showcasing the least successful experiment of number sorting, which utilizes the 20th entry from the dataset (sorting_256.csv file we uploaded). A total of 16 nodes were utilized, numbered from 0 to 14, with the final node included.
>
> The accuracy for each node is as follows:
>
> node 0,1,2 (Top-level nodes): 80.60%, 76.19%, 75.42%
>
> node 3,4,5,6,7,8 (Mid-level nodes): 31.39%, 81.91%, 83.10%, 72.70%, 76.14%, 12.14%
>
> node 9,10,11,12,13,14 (Low-level node): 35.71%, 10.12%, 33.67%, 31.18%, 36.67%, 83.51%
>
> Final node (Final result): 34.98%
>
> The failure occurred because LLM sometimes did not fully understand the rationale information.
>
> For example, node 9 produced the Aggregation Node’s rationale: "The sorted data does not accurately reflect the input data, as it is missing unique numbers such as 1417, 1456, and 1480. Additionally, the sorted list incorrectly includes numbers like 1540 and 1560 that are not present in the input list..." This rationale was passed to the final node. Although none of the preceding nodes indicated that 1540 was included in the input data, the final node answered [… , 1417, 1427, 1430, 1430, 1450, 1456, …, 1480, 1489, 1490, 1500, 1500, 1540, 1540, 1600, 1600, 1700, 1700, 1800, 1800, 1900, 1900, 2000, 2000, 2500, 5000] (The sorting_256 data has number as an element in the list from 0 to 1500.). **In other words, LLM didn't utilize all of the rationale information we provided; instead, LLM utilized only some of the rationale.** This is likely due to the LLM had confidence in the knowledge it had learned during training.
>
> The other four experiments resulted in accuracies of 88.19%, 82.33%, 68.37%, and 88.97%.
>
>
>
> > **Weaknesses 1.2.** the gap between EGoT and EGoT*
>
> Thank you for your feedback. Here, we compare the results between EGoT and EGoT* from the experiment using the first data entry from the sorting_256.csv file. Both architectures propagate rationale in the same manner, leading to similar rationales. However, **the performance of the LLM's response on the Answering Node differs significantly between the two architectures.**
>
> A total of 16 nodes were utilized, numbered from 0 to 14, with the final node included.
>
> EGoT
>
> node 0,1,2 (Top-level nodes): 89.47%, 92.96%, 84.76%
>
> node 3,4,5,6,7,8 (Mid-level nodes): 89.96%, 92.77%, 91.01%, 91.42%, 92.91%, 92.77%
>
> node 9,10,11,12,13,14 (Low-level node): 92.45%, 92.13%, 93.56%, 90.94%, 91.76%, 92.50%
>
> Final node (Final result): 94.34%
>
> EGoT*
>
> node 0,1,2 (Top-level nodes): 86.84%, 94.34%, 96.96%
>
> node 3,4,5,6,7,8 (Mid-level nodes): 88.15%, **33.72%**, 87.32%, 88.24%, 87.27%, 88.85%
>
> node 9,10,11,12,13,14 (Low-level node): 89.30%, 90.67%, 92.08%, 92.86%, 80.80%, 88.60%
>
> Final node (Final result): 91.35%
>
>
> The above result demonstrates that high temperatures can lead to a significant drop in accuracy, as seen in the 33.72% accuracy of node 4. This phenomenon, where the performance drops significantly in the subsequent node and eventually declines further in the final node, is common in EGoT*. The results of the other four trials were as follows:
>
> EGoT: 94.70%, 93.93%, 93.61%, 93.23% vs EGoT*: 93.56%, **49.39%**, 93.25%, 92.50%
>
> We also observed that EGoT* sometimes produces unexpected results, such as 49.39%, even in the final outcome.

---

> ### Author Response · Authors · 2024-11-18
> **Response to Reviewer Gedu (2/3)**
>
> > **Weaknesses 2.** Experiments were limited to GPT-4o models. Findings might differ from other model types.
>
> **Thank you for your valuable suggestion. We ran experiments with Llama 3.1 405B and Mistral 8 * 22B via Fireworks.**
>
> - Llama 3.1 405B
>
> | structure | EGoT (ours) | GoT | ToT | CoT |
> | --- | --- | --- | --- | --- |
> | Accuracy | **95.85%** | 94.09% | 92.05% | 91.59% |
> | Number of Errors | **11.5** | 16.4 | 21.3 | 22.53 |
>
> - Mistral 8 * 22B
>
> | structure | EGoT (ours) | GoT | ToT | CoT |
> | --- | --- | --- | --- | --- |
> | Accuracy | **89.05%** | 83.85% | 71.91% | 82.91% |
> | Number of Errors | **30.67** | 44.6 | 83.6 | 73.63 |
>
> - Anthropic claude-3-haiku
>
> | structure | EGoT (ours) | GoT | ToT | CoT |
> | --- | --- | --- | --- | --- |
> | Accuracy | 95.00% | 94.38% | **97.62%** | 92.10% |
> | Number of Errors | 12.9 | 14.6 | **6.2** | 20.4 |
>
> We used the fireworks(https://fireworks.ai/) platform to conduct experiments for open-sourced LLMs. Also, we experimented with Anthropic’s models, a commercial LLM. However, Anthropic does not provide logprob functionality, so we fixed logprob to 1. Kindly note that the other two models (Mistral, Llama) provide the logprob of their answers.
> We ran the experiment with a total of 10 sorting data and the results are shown in the table. During the experiment, Llama and Mistral, unlike GPT-4o mini, almost always answered score as 100 in the Evaluation Node, in which case we asked score one more time regardless of logprob. The results show that our idea EGoT was the most effective on both Llama and Mistral, and it is slightly effective on Anthropic without logprob.
>
> ---
>
> > **Weaknesses 3.** How the aggregate rationale node functions is unclear. How effectively can LLMs aggregate rationales?
>
> **We have attached some of the prompts and results in the appendix.**
>
> In most cases, if LLM performance is high enough, it can effectively aggregate the rationale well.
> The aggregate rationale node focuses on the negative information to prevent incorrect answers for the next answering node. Sometimes it misunderstands the rationale inputs and aggregates incorrectly. For example, LLM cannot understand the difference between evaluation score and sorting result, so it includes the evaluation score in the sorting result. However, this misinformation disappeared as the graph progressed, so the negative effect was negligible. We have attached some of the prompts and results in the appendix.
>
> The answering node rationale utilizes a portion of the aggregated rationale provided by the upper nodes to generate its response. The evaluation node rationale identifies anomalies in the response, while the aggregate node rationale synthesizes all inputs to produce the final aggregated rationale.
>
> Below is an example of the rationale from a node in the first experiment using the sorting_256.csv dataset:
>
> Answering node rationale: "… However, discrepancies were found in the sorted output, such as the number '1499' which does not exist in the original list, and the number '6' which was also incorrectly included. Additionally, the number '1496' appeared twice in the original list but only once in the sorted output. The number '1098' was also misrepresented, appearing only twice in the sorted list instead of three times as in the original input. Therefore, the sorting process needs to be reevaluated to ensure accuracy and completeness. The final sorted list, after correcting these discrepancies, is: [6, …, 1090, 1098, 1098, 1098, 1111, …, 1496, 1496, 1499]",
>
> Evaluation node rationale: "Upon re-evaluation, the sorted list contains numerous inaccuracies. … For example, the number '582' appears in the sorted data but is not in the input data, while '247' is missing. …",
>
> Aggregate node rationale: "The evaluation of the sorted list reveals significant discrepancies when compared to the original list. Missing elements such as '247' and the presence of extraneous numbers like '582' and '1499' indicate that the sorting process was flawed. Additionally, the number '1496' appears incorrectly in the sorted output, as it is listed only once instead of twice, and '1098' is misrepresented, appearing only twice instead of three times. These issues necessitate a thorough re-evaluation of the sorting process to ensure accuracy and completeness.", "The numbers '582', '1499', '247', '1496', and '1098' require rechecking due to discrepancies in their presence and frequency between the original and sorted lists."
>
> The answering node addressed the rationale it received from the parent node, specifically checking '1499,' '6,' '1496,' and '1098.'
>
> When compared to the correct answer, the LLM's response omitted the numbers “117, 124, 302, 572, 602, 615, 627, 1102, 1108” and included extraneous numbers “134, 145, 247, 582, 682, 996, 1048, 1189.” While the evaluation node failed to provide a fully accurate response, it correctly identified anomalies such as '582' and '247.'"
>
> ---

---

> ### Author Response · Authors · 2024-11-18
> **Response to Reviewer Gedu (3/3)**
>
> > **Questions 1.** Given that EGoT requires three times the resources compared to other mechanisms, for which tasks does this approach offer an advantage, and in which cases might it be less suitable?
>
> Our research highlights the advantages of **incorporating rationale information into prompt engineering**, instead of focusing only on LLM's answer. We propose a structured approach to enhance prompt engineering performance by **leveraging the impact of the LLM's temperature**.
>
> The prompts provided in src/prompt/human are designed for ease of use and operate without requiring human interaction to produce the final result. Unlike other methods that include detailed examples in their prompts, ours intentionally avoids providing specific examples to minimize bias in the results [1], [2]. Notably, while GoT's capabilities are limited to solving partitionable large-scale problems, our approach can address non-partitionable large problems by leveraging rationale (e.g., frozen lake problem).
>
> ---
>
> [1] https://github.com/princeton-nlp/tree-of-thought-llm/blob/master/src/tot/prompts/game24.py#L3
>
> [2] https://github.com/spcl/graph-of-thoughts/blob/main/examples/sorting/sorting_128.py#L134
>
> ---
>
> > **Questions 2.** Is there any rationale behind selecting these specific experimental setups - document merging, number sorting, and frozen lake?
>
> We chose these specific experimental setups to evaluate our approach in diverse contexts and benchmark it against existing methods. The sorting and document merging problems were selected because they are part of the GoT structure, which represents the latest state-of-the-art (SOTA) in this area, allowing for a direct comparison. **To explore a more complex and well-known problem**, we included the Frozen Lake scenario, a classic example in reinforcement learning (RL). This choice aligns with our approach’s focus on leveraging rationale, drawing exploration versus exploitation: higher temperatures facilitate exploration, while lower temperatures encourage exploitation.

---

> ### Author Response · Authors · 2024-11-25
> **Official Comment by Authors**
>
> Dear Reviewer Gedu,
>
> I hope this message finds you well. As the end of the discussion period is approaching, we have not yet received any feedback. We have made significant efforts to address the concerns you raised during the initial review, and we would be grateful for any further questions or feedback you may have. Additionally, we would be grateful if you could kindly share whether our revision has impacted your evaluation.
> Thank you very much for your time and valuable insights. We look forward to hearing from you soon.
>
> Sincerely,
>
> Authors of Submission6264

---

> ### Author Response · Authors · 2024-12-01
> **Official Comment by Authors**
>
> Dear Reviewer Gedu,
>
> I hope this message finds you well. As the end of the extended discussion period is approaching, we have not yet received any feedback. We have made significant efforts to address the concerns you raised during the initial review, and we would be grateful for any further questions or feedback you may have. Additionally, we would be grateful if you could kindly share whether our revision has impacted your evaluation. Thank you very much for your time and valuable insights. We look forward to hearing from you soon.
>
> We have made the following revisions to the paper in response to the concerns raised by Reviewer Gedu. For a detailed overview of the changes, we kindly invite you to refer to our official comments.
>
> **W1**: We have added Section 5.4.
>
> **W2**: We have included content from lines 372 to 377 in Section 5.1 and updated Table 2.
>
> **W3**: We have added examples in the appendix.
>
> **Q1**: We have added content on lines 66-67.
>
> Sincerely,
>
> Authors of Submission6264

---

### Author Response · Authors · 2024-11-28
**Gentle Reminder: Deadline for PDF update**

Dear Reviewers,

Thank you once again for your invaluable comments and insightful feedback on our submission. We truly appreciate your time and effort in reviewing our work and providing such detailed guidance.

As the deadline for submitting the updated manuscript approaches, we kindly ask if you have any additional feedback or concerns regarding our recent revisions. Your feedback on both the updated content and the evaluation would be incredibly helpful in ensuring we address all points thoroughly before submitting the updated version.

We deeply value your support and are committed to making the necessary improvements. Given the extension of the Discussion Period, we kindly request that you share any further suggestions or thoughts that could help enhance the quality of our submission. Your additional feedback would be greatly appreciated.

Sincerely,

Authors of Submission6264

---

### Author Response · Authors · 2024-11-28
**Major Changes in the Manuscript**

Dear Reviewers,

I would like to express my sincere gratitude for your valuable time and effort in reviewing our manuscript. We greatly appreciate the insightful feedback you provided. Based on your comments, we have made the following revisions:

Major Changes

Section 1, lines 65-67: We have added this section to highlight the novelty of our approach.

Section 2, lines 83-88: We have included additional content to provide a clearer and more comprehensive overview of EGoT.

Section 5.1, lines 372-377, Table 2: We have added additional experimental results using other LLM models to further support our findings.

Section 5.3, lines 426-428: We have emphasized the advantages of our method.

Section 5.4: We have included a discussion of EGoT's strengths, cases where it might fail, and a comparison between EGoT and EGoT*.

Section 7, lines 503-504: We have clarified the distinction between our method and fine-tuning.

Appendix A.1: We have added details related to costs.

Appendix A.2 - A.5: We have included the prompts used in our experiments, and examples of the experimental results.

We hope these changes enhance the quality of the manuscript. Once again, we deeply appreciate your valuable comments and suggestions.

Sincerely,

Authors of Submission6264

---

### Meta-Review · Area_Chair_nkir · 2024-12-19

**Metareview:**

**Summary**

This paper proposes Enhancing Graph of Thoughts (EGOT) which is a dynamic an approach of enhancing prompts with dynamic temperature control.They design a flow within each node to answer the question with rationales, evaluate the answer with rationales, and consolidate the answer and evaluation rationales as inputs to the child nodes. They also introduce temperature control with cosine annealing to explore answer space at earlier nodes and refine the promising answers at later nodes. They report that the proposed methodology shows better performance on number sorting and frozen lake tasks compared to baselines including CoT, ToT, and GoT.


**Strengths**

The paper introduces a novel prompting strategy: graph-of-thoughts that seems to outperform existing strategies in solvin tasks

**Weaknesses**

**Final remarks**

The paper greatly improved during the interaction although some reviewers have not answered.

**Additional Comments On Reviewer Discussion:**

During the discussion, additional experiments have been carried out.

---

### Decision · Program_Chairs · 2025-01-22

Accept (Poster)